# Safety and Efficacy of the LVIS EVO Device for Stent-Assisted Coiling of Intracranial Aneurysms: A Systematic Review and Meta-Analysis

**DOI:** 10.3390/jcm15010260

**Published:** 2025-12-29

**Authors:** Abdulrahim Saleh Alrasheed, Amna Mutasim Elazrag, Rola Hamad Alseghair, Majd Nouh Alasmari, Mohammad Salem Alqahtani, Hosam Al-Jehani

**Affiliations:** 1Department of Neurosurgery, College of Medicine, King Faisal University, Al Ahsa 31982, Saudi Arabia; 2Faculty of Medicine, University of Khartoum, Khartoum 11115, Sudan; 3College of Medicine, Qassim University, Qassim 51452, Saudi Arabia; 4College of Medicine, King Khalid University, Abha 61421, Saudi Arabia; 5Neurosurgery Department, King Fahad Hospital of the University, Imam Abdulrahman Bin Faisal University, Khobar 31451, Saudi Arabia; 6Department of Neurosurgery, Interventional Neuroradiology and Critical Care Medicine, Imam Abdulrahman Bin Faisal University, Khobar 31451, Saudi Arabia; 7Department of Neurology and Neurosurgery, Montreal Neurological Institute, McGill University Health Centre, Montreal, QC H4A 0B1, Canada; 8Department of Neurosurgery, Weill Cornell University, Houston Methodist, Houston, TX 77030, USA

**Keywords:** intracranial aneurysms, stent-assisted coiling, endovascular treatment, neurointervention

## Abstract

**Background**: The low-profile visualized intraluminal support EVO (LVIS EVO) stent is a novel device for stent-assisted coiling (SAC) that offers improved visibility, radial force, and navigability in tortuous cerebral vessels. This systematic review and meta-analysis evaluated the safety and efficacy of the LVIS EVO stent in treating intracranial aneurysms (IAs). **Methods**: A systematic search of PubMed, Web of Science, Cochrane Library, and Scopus databases to identify studies reporting clinical outcomes of LVIS EVO in SAC of IAs was performed. Primary outcomes included aneurysm occlusion based on the Raymond–Roy Occlusion Classification (RROC), procedural, and clinical complication rates. Pooled estimates with 95% confidence intervals (CIs) were calculated using either a fixed- or random-effects model. Heterogeneity was assessed using the I^2^ statistic. Sensitivity analyses were conducted using the leave-one-out method. **Results**: Twelve studies involving 567 patients and 586 aneurysms were included. Immediate angiographic results showed complete occlusion (RROC I) in 78% (95% CI: 0.53–0.92), residual neck (RROC II) in 13% (95% CI: 0.08–0.21), and residual aneurysm (RROC III) in 9% (95% CI: 0.02–0.34). At follow-up, complete obliteration was observed in 73% (95% CI: 0.60–0.83). Procedural complications occurred in 5% (95% CI: 0.03–0.09), and clinical complications in 6% (95% CI: 0.04–0.10). **Conclusions**: The LVIS EVO stent demonstrated high technical success, effective aneurysm occlusion, and low complication rates, supporting its safety and efficacy. Prospective studies are warranted to confirm long-term outcomes.

## 1. Introduction

Intracranial aneurysms (IAs) are localized dilatations of cerebral arteries that arise due to structural weaknesses in the arterial wall, often at branching points of major vessels within the circle of Willis. Almost 2–3.2% of the global population have IAs. They may remain clinically silent or present with nonspecific symptoms such as persistent headaches, visual disturbances, nausea, vomiting, or cranial nerve palsies. However, their most catastrophic presentation is rupture, leading to subarachnoid hemorrhage (SAH), a neurological emergency associated with high morbidity and mortality. Approximately 15% of patients die before reaching medical care, contributing to an overall case-fatality rate approaching 50%. Therefore, early diagnosis and management are paramount [1].

Wide-necked and complex IAs pose additional management challenges due to the technical difficulty in achieving stable aneurysm occlusion and the higher risk of procedural complications such as coil prolapse or incomplete embolization [2,3]. Various devices and techniques have been introduced to overcome the technical limitations associated with treating wide-necked IAs, including three-dimensional coils, multiple microcatheters, balloon remodeling technique, intracranial stents, flow diverter devices, and neck-bridge devices [3].

Stent-assisted coiling (SAC) is a common endovascular procedure for wide-necked and complex IAs. It provides a scaffold across the aneurysm neck, improving coil retention, stability, and long-term occlusion rates [4,5,6,7,8]. Among the stents used in SAC, self-expanding nitinol stents such as Neuroform Atlas [9] and LVIS Jr [10] offer flexibility and navigability, allowing safe deployment in tortuous cerebral vessels. A notable advancement in the category of self-expanding stents is the low-profile visible intraluminal support EVO (LVIS EVO) (MicroVention, Terumo, Aliso Viejo, CA, USA), a second-generation low-profile visualized intraluminal support device [11].

The LVIS EVO stent has emerged as a next-generation device offering enhanced radiopacity and improved intra-aneurysmal flow disruption. These features allow it to function as a hybrid stent between traditional SAC and flow diversion, especially in anatomically challenging IAs [12,13]. Although originally designed as a braided assist device, it has been utilized in selective studies as a flow diverter due to its high metal coverage and flow-modifying properties, particularly in small or distal IAs [12]. Although there is a growing adoption of the LVIS EVO stent in IA management, there is no comprehensive study of its safety, efficacy, and feasibility in real clinical practice. This study sought to assess these aspects in real clinical settings.

## 2. Methods

We followed the preferred reporting items for systematic reviews and meta-analyses (PRISMA 2020; Appendix A) guidelines [14] in conducting our study (Online Resource 1). The protocol for this review was registered prospectively in the PROSPERO database (CRD42024621191).

### 2.1. Search Strategy

A comprehensive literature search was performed across electronic databases, PubMed, Scopus, Web of Science, and Cochrane Library, through December 2024. The search strategy included a combination of medical subject headings (MeSH) terms and free-text keywords: (LVIS EVO OR LVIS OR Low Profile Visible Intraluminal Support EVO OR EVO OR Low Profile Visible Intraluminal Support OR Stent-Assisted Coiling) AND (Intracranial Aneurysm OR Brain Aneurysm OR Cerebral Aneurysm). The reference lists of the included articles were manually screened to identify additional eligible articles.

### 2.2. Inclusion and Exclusion Criteria

Eligible studies included adults (≥18 years) diagnosed with IAs that were treated using LVIS EVO stents, assessing either safety or efficacy, regardless of comparison to alternative treatments. Exclusion criteria included pediatric population, conference articles, editorials, case reports, and reviews.

### 2.3. Study Selection

Two independent reviewers conducted this process using Rayyan software v1.5.6 [15]. First, titles and abstracts were screened to exclude irrelevant articles based on the predefined eligibility criteria and duplicate record removal. Subsequently, full-text screening of potentially eligible studies was performed. Any conflicts were resolved through discussion. Our initial search yielded a total of 2542 records. After removing duplicates, 1895 records remained. Following title and abstract screening, 1391 records were eliminated. Subsequently, 504 full-text articles were assessed for eligibility. Ultimately, only 12 articles met the inclusion criteria (Figure 1).

### 2.4. Data Extraction

A standardized data extraction sheet was used. The following variables were recorded: study characteristics (e.g., first author’s name, publication year, country, study design, and study period), patient demographics (e.g., age, gender, and diagnosis), aneurysm characteristics (e.g., location, size, and type), and intervention details (e.g., treatment indication, previous treatments, and adjunctive medication use).

### 2.5. Outcome Measures

The primary outcome measures were the immediate technical success of stent deployment and the degree of aneurysm occlusion based on the Raymond–Roy Occlusion Classification (RROC) that classifies coiled IA into 3 classes. Class I reflects complete occlusion, class II reflects residual neck, and class III reflects residual aneurysm. Modified RROC classes IIIA and IIIB were pooled as class III. Secondary outcomes included procedure-related and clinical complications.

### 2.6. Quality Assessment

Two independent reviewers conducted the quality assessments. Any disagreements in scoring were resolved through discussion. The Newcastle–Ottawa Scale (NOS) for observational studies was utilized. The NOS was modified by removing the “comparability” and “selection of non-exposed group” due to the single-arm study’s nature [16]. Their ratings were used to inform the interpretation and reliability of results.

### 2.7. Data Analysis

Pooled estimates were conducted using Review Manager (RevMan) software, v5.4 [17]. Heterogeneity among studies was assessed through visual inspection of funnel plots and the I^2^ statistic. A fixed-effect model was considered to be used when I^2^ was <40%. For analyses with substantial heterogeneity (I^2^ ≥ 40%), a random-effects model was applied. To explore heterogeneity sources, leave-one-out sensitivity analyses were conducted. Publication bias was assessed through funnel plot visualization. All statistical tests were two-sided, with *p*-values < 0.05 considered statistically significant.

## 3. Results

### 3.1. Study Characteristics

A total of 12 studies, encompassing 567 patients and 586 aneurysms, were included in this review, all employing a retrospective study design. Three originated from Germany [13,18,19], two from Australia [11,20], and one each from Poland [21], Bulgaria [22], Turkey [23], and the United States [24]. Additionally, two studies were multicenter collaborations, one involving 11 European centers [12] and another including centers from Turkey, the United Kingdom, and Kazakhstan [25]. Females represented the majority of the included patients, accounting for 401 females (70.7% of the total), with mean ages ranging from 53 [22] to 63 years [11]. Several studies reported the percentage of patients who had undergone previous treatments. Poncyljusz et al. (2020) noted that 13.3% had received endovascular coil embolization [21], while Vollherbst et al. (2021) reported that 15.3% had prior coiling or other treatments [12]. Other studies highlighted balloon-assisted coiling, intraluminal flow disruption, or surgical treatment as previous interventions (Table 1).

**Table 1 jcm-15-00260-t001:** General characteristics of the included studies.

Author	Country	Study Design	Previous Treatment, N (%)	Treatment Indications or Clinical Presentation, N (%)	Age (Years), Mean (SD)	Male, N (%)	Female, N (%)
Poncyljusz et al., 2020 [21]	Poland	Retrospective study	4 (13.3%) Endovascular coil embolization	Acute intracranial hemorrhage due to aneurysm rupture; elective unruptured IAs	60.76 (18.1)	6 (20%)	24 (80%)
Vollherbst et al., 2021 [12]	Various European countries	Retrospective study	7 (11.9%) Endovascular coil embolization; 2 (3.4%) Previous treatment	34 (57.6%) Incidental IAs; 7 (11.9%) Recanalization; 9 (15.3%) SAH; 5 (8.5%) Symptomatic IAs; 2 (3.4%) Ischemic stroke; 2 (3.4%) Residual IAs	58.5 (12)	15 (26.3%)	42 (73.7%)
Sirakov et al., 2020 [22]	Bulgaria	Retrospective study	3 (50%) Endovascular coil embolization	Wide-necked or recurrent IAs	53 (9.8)	3 (50%)	3 (50%)
Maus et al., 2021 [18]	Germany	Retrospective study	4 (26.7%) Endovascular coil embolization	Wide-necked IAs	56 (7.7)	7 (47%)	8 (53%)
Islak et al., 2025 [23]	Turkey	Retrospective study	2 (5.4%) Balloon-assisted coiling	34 (91.9%) Elective unruptured IAs; 2 (5.4%) Sudden severe headache without confirmed SAH; 1 (2.7%) Ruptured IAs	53.9 (11.5)	10 (27%)	27 (73%)
Aydin et al., 2023 [25]	Turkey, UK, Kazakhstan	Retrospective study	22 (21.4%) Coiling or implantation of instrasaccular flow disruptor	Wide-necked complex or recanalized IAs	54.9 (11.3)	40 (38.8%)	63 (61.2%)
Kayan et al., 2025 [24]	USA	Retrospective study	Most of the patients underwent SAC	NA	62.7 (16.7)	13 (24.5%)	40 (75.5%)
Settipalli et al., 2024 [20]	Australia	Retrospective study	NA	Elective unruptured IAs	58 (10.9)	8 (30.7%)	18 (69.23%)
Maurer et al., 2023 [13]	Germany	Retrospective study	7 (5.9%) Endovascular treatment; 3 (2.5%) Surgical treatment	Wide-necked IAs	57.3 (7.3)	30 (26.8%)	82 (73.2%)
Kubiak et al., 2023 [26]	NA	Retrospective study	NA	Ruptured blood blister-like IAs	55.3 (10.7)	5 (50%)	5 (50%)
Foo et al., 2021 [11]	Australia	Retrospective study	1 (6.7%) Previous WEB device	6 (40%) Elective surveillance; 1 (6.7%) Elective surveillance with left CN6 palsy; 1 (6.7%) Elective surveillance with left partial hemianopia; 1 (6.7%) Incidental IAs; 1 (6.7%); Elective surveillance with right visual impairment; 1 (6.7%) Elective surveillance with SAH; 3 (20%); Emergency indication due to SAH; 1 (6.7%) Emergency indication due to headache	63 (11.8)	5 (33.3%)	10 (66.7%)
Mosimann et al., 2022 [19]	Germany	Retrospective study	24 (23%) Previous coiling; 2 (2%); Previous clipping	Unruptured intracranial saccular aneurysms	53.3 (12.4)	24 (23%)	79 (77%)

NA: Not Applicable; IAs: Intracranial Aneurysms; SAH: Subarachnoid Hemorrhage; SAC: Stent-Assisted Coiling; CN6: Cranial Nerve 6; WEB: Woven EndoBridge.

### 3.2. Aneurysm’s Characteristics

Most of the IAs were located in the middle cerebral artery (MCA) (34.8%), followed by the anterior communicating artery (AComA) (19.6%), the internal carotid artery (ICA) (15.9%), and the basilar artery (11.8%). Anterior cerebral artery (ACA) aneurysms accounted for 10.2% of the total. The remaining IAs (7.7%) were located in other various regions. Regarding the aneurysm status, most of the aneurysms were incidental IAs (32.6%). Recanalized aneurysms were reported in 9.9%, while ruptured aneurysms were documented in 9%. Saccular aneurysms were the most commonly reported morphological type across multiple studies, including Sirakov et al. (2020) [22], Maus et al. (2021) [18], and Islak et al. (2025) [23], where saccular morphology accounted for 100% of cases. Some studies noted fewer common types, such as blister-like or dissecting aneurysms (Table 2).

**Table 2 jcm-15-00260-t002:** Aneurysm’s Characteristics.

Author	Aneurysm Location, N (%)	Aneurysm Presentation, N (%)	Aneurysm Morphology, N (%)
Poncyljusz et al., 2020 [21]	15 (42.9%) ICA; 11 (31.4%) MCA; 4 (11.4%) AComA; 4 (11.4%) BA; 1 (2.9%) ACA	25 (71.4%) Incidental IAs; 4 (11.4%) Recanalized IAs; 6 (17.1%) Ruptured IAs	NA
Vollherbst et al., 2021 [12]	7 (11.9%) ICA; 15 (25.4%) MCA; 13 (22%) AComA; 9 (15.3%) BA; 6 (10.2%) ACA; 9 (15.3%) Other	34 (57.6%) Incidental IAs; 7 (11.9%) Recanalized IAs; 9 (15.3%) SAH; 5 (8.5%) Symptomatic IAs; 2 (3.4%) Ischemic stroke; 2 (3.4%) Residual IAs	55 (93.2%) Saccular; 1 (1.7%) Blister-like; 3 (5.1%) Dissecting
Sirakov et al., 2020 [22]	3 (50%) MCA; 1 (16.7%) AComA; 1 (16.7%) BA; 1 (16.7%) ACA	3 (50%) Incidental IAs; 3 (50%) Recanalized IAs	6 (100%) Saccular
Maus et al., 2021 [18]	1 (6.7%) ICA; 9 (60%) MCA; 4 (26.7%) AComA; 1 (6.7%) ACA	4 (26.7%) Recanalized IAs; 2 (13.3%) Ruptured IAs	15 (100%) Saccular
Islak et al., 2025 [23]	23 (62%) MCA; 5 (13.5%) AComA; 7 (19%) BA; 2 (5.4%) PComA	2 (5.4%) Incidental IAs; 2 (5.4%) Recanalized IAs; 1 (2.7%) SAH; 28 (80%) Headache; 1 (2.7%) Unilateral numbness; 1 (2.7%) Hemiparesis; 1 (2.7%) Vertigo; 1 (2.7%) Gait and balance deficit	37 (100%) Saccular
Aydin et al., 2023 [25]	37 (35.9%) MCA; 31 (30.1%) AComA; 14 (13.6%) BA; 3 (2.9%) ACA; 8 (7.8%) Carotid artery—Ophthalmic segment; 7 (6.8%) Carotid artery—bifurcation; 2 (1.9%) SCA; 1 (1%) Carotid artery—communicating segment	30 (29.1%) Incidental IAs; 2 (1.9%) Recanalized IAs; 19 (18.4%) Recurrence; 41 (39.8%) Headache; 4 (3.9%) Screening with a family history; 7 (6.8%) Other symptoms	NA
Kayan et al., 2025 [24]	1 (2%) ICA terminus; 17 (31%) MCA; 20 (35%) AComA; 8 (14%) BA; 3 (5%) ACA; 3 (5%) PComA; 2 (3%) PICA; distal MCA 1 (2%)	NA	NA
Settipalli et al., 2024 [20]	4 (14.8%) BA; 18 (66.7%) AComA; 4 (14.8%) ACA; 1 (3.7%) ICA	NA	27 (100%) Saccular
Maurer et al., 2023 [13]	27 (22.9%) ICA; 44 (37.3%) MCA; 6 (5.1%) BA; 36 (30.5%) ACA; 2 (1.7%) PCA; 2 (1.7%) PICA; 1 (0.8%) VA	97 (82.2%) Incidental IAs; 15 (12.7%) Acute SAH; 2 (1.7%) Acute cranial nerve palsy; 2 (1.7%) Acute stroke; 1 (0.8%) Seizure; 10 (8.5%) Residual IAs	NA
Kubiak et al., 2023 [26]	11 (84.6%) ICA; 1 (7.7%) MCA; 1 (7.7%) VA	13 (100%) Ruptured IAs	13 (100%) Blister-like
Foo et al., 2021 [11]	4 (26.6%) ICA; 5 (33.3%) AComA; 2 (13.3%) BA; 1 (6.6%) ACA; 1 (6.6%) PICA; 1 (6.6%) PCA; 1 (6.6%) PComA	3 (20%) Ruptured IAs; 12 (80%) Unruptured IAs	1 (6.6%) Blister-like; 12 (80%) Single lobe; 1 (6.6%) Bilobed; 1 (6.6%) Multi-lobed
Mosimann et al., 2022 [19]	25 (24.3%) ICA; 42 (40.8%) MCA; 14 (14%) AComA; 14 (14%) BA; 4 (3.9%) ACA; 2 (2%) PICA; 2 (2%) SCA	26 (25%) Recanalized IAs; 77 (75%) Untreated IAs	103 (100%) Saccular

IAs: Intracranial Aneurysms; ICA: Internal Carotid Artery; MCA: Middle Cerebral Artery; AcomA: Anterior Communicating Artery; BA: Basilar Artery; ACA: Anterior Cerebral Artery; PcomA: Posterior Communicating Artery; PICA: Posterior Inferior Cerebellar Artery; SCA: Superior Cerebellar Artery; PCA: Posterior Cerebral Artery; SAH: Subarachnoid Hemorrhage; VA: Vertebral Artery.

### 3.3. Quality Assessment

Overall, the studies showed a low to moderate risk of bias. However, the follow-up domain showed some limitations, reflecting the challenges of retrospective studies in ensuring adequate follow-up and monitoring periods (Table 3).

**Table 3 jcm-15-00260-t003:** Risk of Bias Assessment of the Included Studies Using NOS.

Author	Representativeness of the Exposed Cohort (★)	Ascertainment of Exposure (★)	Demonstration That Outcome of Interest Was Not Present at Start of Study (★)	Assessment of Outcome (★)	Was Follow-Up Long Enough for Outcomes to Occur? (★)	Adequacy of Follow-Up of Cohorts (★)	Quality Level
Poncyljusz et al., 2020 [21]	*	*	*	*	0	0	4
Vollherbst et al., 2021 [12]	*	*	*	*	0	0	4
Sirakov et al., 2020 [22]	*	*	*	*	*	*	6
Maus et al., 2021 [18]	*	*	*	*	0	0	4
Islak et al., 2025 [23]	*	*	*	*	*	*	6
Aydin et al., 2023 [25]	*	*	*	*	*	*	6
Kayan et al., 2025 [24]	*	*	*	*	*	*	6
Settipalli et al., 2024 [20]	*	*	*	*	0	0	4
Maurer et al., 2023 [13]	*	*	*	*	*	*	6
Kubiak et al., 2023 [26]	*	*	*	*	0	0	4
Foo et al., 2021 [11]	*	*	*	*	0	0	4
Mosimann et al., 2022 [19]	*	*	*	*	*	*	6

★ Indicated a point for each question if the study aligned with the question, and 0 indicated no point if the study failed to align with the question. A total score of 5–6 stars is considered a low risk of bias, 3–4 stars indicate a moderate risk, and 1–2 stars indicate a high risk of bias.

### 3.4. Immediate RROC

The pooled proportions for ROCC I, II, and III were 0.78 (95% CI: 0.53–0.92), 0.13 (95% CI: 0.08–0.21), and 0.09 (95% CI: 0.02–0.34), respectively (Figure 2, Figure 3 and Figure 4).

**Figure 2 jcm-15-00260-f002:**
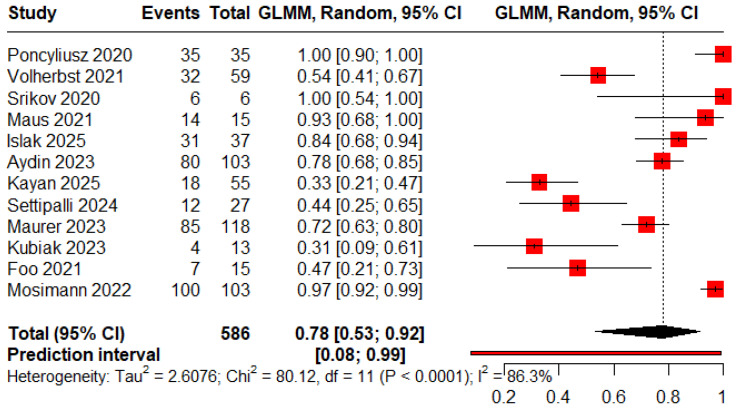
Forest plot showing the proportion of patients achieving immediate complete obliteration [11,12,13,18,19,20,21,22,23,24,25,26].

**Figure 3 jcm-15-00260-f003:**
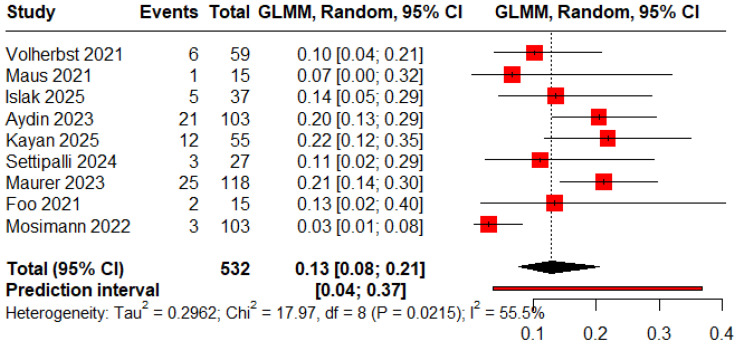
Forest plot showing the proportion of patients with an immediate residual neck [11,12,13,18,19,20,23,24,25].

**Figure 4 jcm-15-00260-f004:**
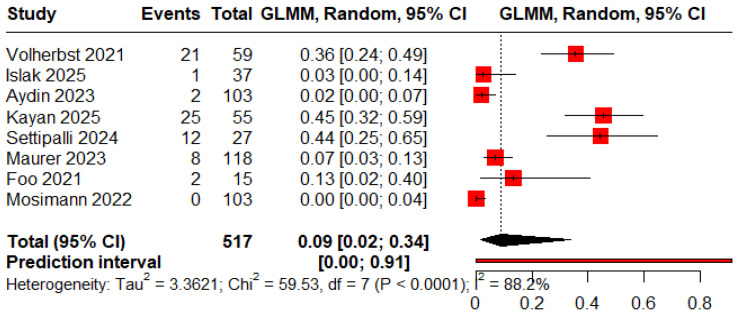
Forest plot showing the proportion of patients with an immediate residual aneurysm [11,12,13,19,20,23,24].

### 3.5. Follow-Up RROC

The pooled proportions for ROCC I, II, and III were 0.73 (95% CI: 0.60–0.83), 0.09 (95% CI: 0.04–0.18), and 0.05 (95% CI: 0.03–0.09), respectively (Figure 5, Figure 6 and Figure 7).

**Figure 5 jcm-15-00260-f005:**
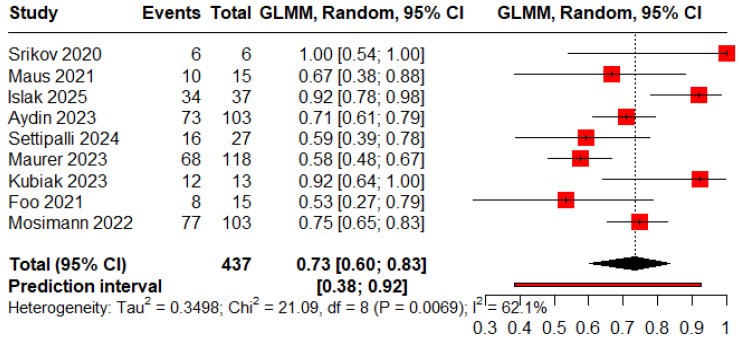
Forest plot showing the proportion of patients achieving complete obliteration during the follow-up period [11,13,18,19,20,22,23,25,26].

**Figure 6 jcm-15-00260-f006:**
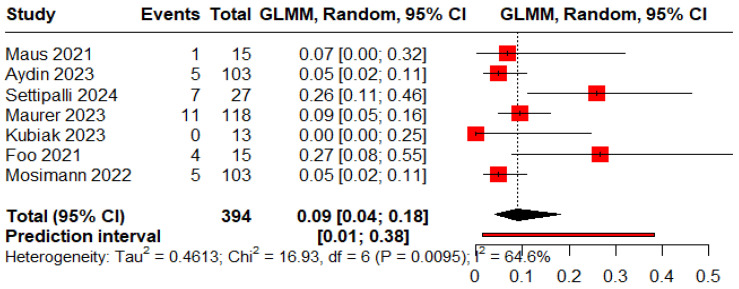
Forest plot showing the proportion of patients with a residual neck during the follow-up period [11,13,18,19,20,25,26].

**Figure 7 jcm-15-00260-f007:**
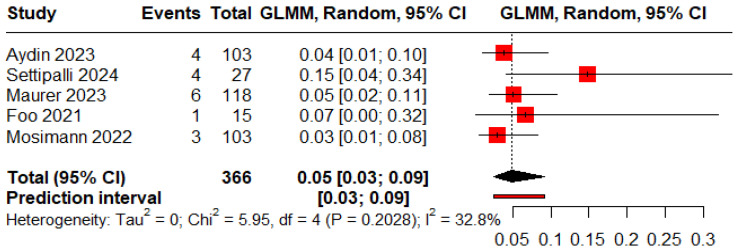
Forest plot showing the proportion of patients with a residual aneurysm during the follow-up period [11,13,19,20,25].

### 3.6. Complications

The pooled proportions of procedure-related and clinical complications were 0.05 (95% CI: 0.03–0.09) and 0.06 (95% CI: 0.04–0.10), respectively (Figure 8 and Figure 9).

**Figure 8 jcm-15-00260-f008:**
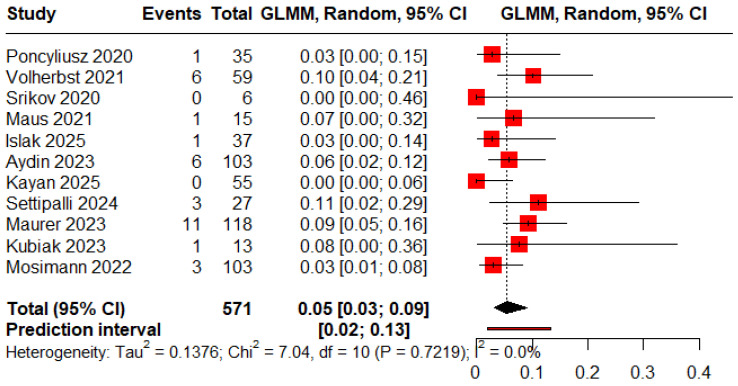
Forest plot of rate of procedure-related complications [12,13,18,19,20,21,22,23,24,25,26].

**Figure 9 jcm-15-00260-f009:**
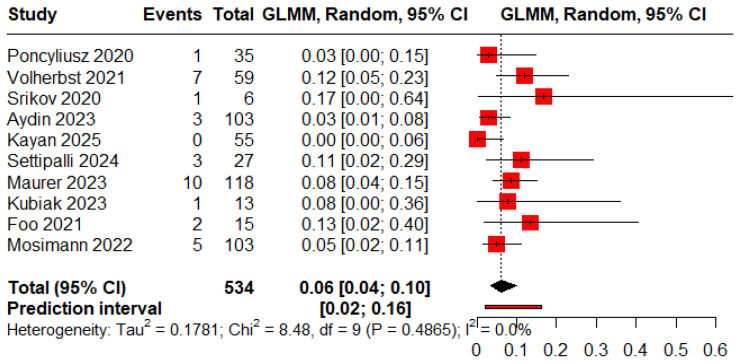
Forest plot of rate of clinical complications [11,12,13,19,20,21,22,24,25,26].

### 3.7. Complications Summary

The overall trend was favorable, as most of the studies reported either a relatively low incidence of complications or none at all. Thrombus formation was the most frequently reported procedure-related complication, with an incidence rate ranging from 2% [25] to 10% [26]. Additionally, other technical complications such as stent shortening [12], inadequate stent opening, coil protrusion [20], and guidewire perforation [19] have been reported. Post-procedure complications were generally low but included serious events such as death, accounting for 2.9% in Poncyljusz et al. (2020) [21], 1.8% in Maurer et al. (2023) [13], and 10% in Kubiak et al. (2023) [26] (Table 4).

**Table 4 jcm-15-00260-t004:** Procedural and clinical outcomes.

Author	Procedural Complications, N (%)	Clinical Complications, N (%)
Poncyljusz et al., 2020 [21]	1 (2.9%) Thrombus formation	1 (2.9%) Death
Vollherbst et al., 2021 [12]	3 (5.1%) Thrombus formation; 1 (1.7%) Stent shortening; 1 (1.7%) Incomplete stent opening; 1 (1.7%) Coil protrusion	2 (3.4%) TIA; 1 (1.7%) Major stroke; 1 (1.7%) Minor stroke; 1 (1.7%) GIT bleeding; 1 (1.7%) Leg ischemia; 1 (1.7%) Puncture site bleeding
Sirakov et al., 2020 [22]	0	1 (16.7%) Tingling sensation
Maus et al., 2021 [18]	1 (7%) Isolated fornix infarction	NA
Islak et al., 2025 [23]	1 (2.7%) Thrombus formation	NA
Aydin et al., 2023 [25]	2 (2%) Thrombus formation; 4 (4%) In-stent thrombus	1 (0.97%) Small insular infarct; 1 (0.97%) Basal ganglia infarct; 1(0.97%) Femoral pseudo-aneurysm
Kayan et al., 2025 [24]	0	0
Settipalli et al., 2024 [20]	1 (3.7%) Inadequate stent opening; 1 (3.7%) Coil tail prolapse; 1 (3.7%) SCA branch occlusion	3 (11.1%) Post-procedure arteriotomy; 1 (3.7%) Inadvertent micro wire perforation of the deep circumflex artery; 1 (3.7%) Inadvertent intraluminal deployment of Angioseal; 1 (3.7%) Non-occlusive common femoral vein thrombus formation; 2 (7.4%) Silent thromboembolic infract; 2 (7.4%) New diffusion-weighted imaging lesions on MRI
Maurer et al., 2023 [13]	7 (5.9%) Thromboembolic complications; 4 (3.4%) Target aneurysm rupture	4 (3.4%) SAH with vasospasm; 1 (0.85%) Vision loss; 1 (0.85%) Hemineglect; 2 (1.8%) Death; 1 (0.85%) Third cranial nerve palsy; 1 (0.85%) Left-sided hemiparesis
Kubiak et al., 2023 [26]	1 (10%) In-stent thrombosis	1(10%) Death
Foo et al., 2021 [11]	NA	2 (13.3%) Symptomatic thromboembolic complications
Mosimann et al., 2022 [19]	1 (1.1%) In-stent Thrombosis; 2 (2%) Guidewire perforation	1 (1%) Femoral pseudoaneurysm; 1 (1%) Retroperitoneal hematoma; 1 (1%) Death; 1 (1%) Occlusion of superficial femoral artery; 1 (1%) PICA infarct (transient ataxia)

TIA: Transient Ischemic Attack; GIT: Gastrointestinal Tract; SCA: Superior Cerebellar Artery; MRI: Magnetic Resonance Imaging; SAH: Subarachnoid Hemorrhage; PICA: Posterior Inferior Cerebellar Artery.

### 3.8. Leave-One-Out Sensitivity Analysis

We used sequential leave-one-out sensitivity analyses to assess heterogeneity by excluding one study at a time. For immediate RROC I, RROC II, and RROC III, the pooled estimates were 0.79 (95% CI: 0.31–0.97), 0.17 (95% CI: 0.14–0.22), and 0.16 (95% CI: 0.01–0.73), after outlier exclusion (Appendix A), respectively. For follow-up RROC I and RROC II, the pooled estimates were 0.69 (95% CI: 0.59–0.77) and 0.07 (95% CI: 0.04–0.12), respectively (Appendix A) (Online Resource 1).

### 3.9. Publication Bias Assessment

Funnel plots were used for publication bias assessment. Immediate RROC had asymmetrical distributions with probable small-study effects (Appendix A). Follow-up analyses showed similar asymmetry, with smaller studies dispersed asymmetrically around the mean effect size (Appendix A). Procedural and clinical complication rates displayed asymmetrical funnel plots. Smaller studies were clustered towards higher effect sizes (Appendix A) (Online Resource 1).

## 4. Discussion

### 4.1. Summary of Findings

In this systematic review and meta-analysis, we analyzed 567 patients with 586 aneurysms. Our findings highlight the potential of LVIS EVO stents as a safe and effective tool in the SAC of IAs, particularly for wide-necked and complex cases that have traditionally been challenging to treat. A high technical success rate has been achieved, as it was deployed successfully in the desired positions and maintained stable aneurysm coiling in most cases. Immediate post-procedural aneurysm occlusion rates were promising, with 78% of aneurysms achieving complete occlusion immediately after the procedure. This initial effectiveness was largely sustained over time, with 73% of aneurysms showing complete occlusion at various follow-up periods. These findings underscore the promising role of LVIS EVO stents in the real-world clinical setting.

Overall, the incidence of complications was relatively low. Procedure-related complications occurred in about 5% of cases, with thrombus formation being the most commonly reported issue. Additionally, there were notable variations in clinical complication rates across different studies, likely due to discrepancies in how complications were defined. Clinical complications were observed in approximately 6% of patients, including symptomatic thromboembolic events, neurological deficits, and death. The need for retreatment was reported only in four studies: 2.4% by Aydin et al. (2023) [25], 3.7% Settipalli et al. (2024) [20], 6% Mosimann et al. (2022) [19] and 6.8% Vollherbst et al. (2021) [12], indicating sustained aneurysm occlusion in most patients. Cortez et al. (2025) [27] reported comparable findings in terms of occlusion rates and safety outcomes. However, our study expands on the current knowledge by including safety outcomes subgroup analyses, and a larger patient population, thereby further consolidating the existing evidence.

### 4.2. Clinical and Technical Implications of LVIS EVO in the Management of IAs

The international study of unruptured intracranial aneurysms (ISUIAs) highlighted that aneurysm size, location, and morphology significantly influence rupture risk, with posterior circulation and wide-necked aneurysms posing poorer surgical outcomes [28]. SAC offers a safer and effective treatment option for many patients with ruptured aneurysms, especially those with wide-necked aneurysms [29]. SAC provides scaffold support to prevent coil prolapse, enabling more secure and complete embolization, particularly in wide-neck aneurysms with neck widths >4 mm or dome-to-neck ratios <2 mm [12].

The braided LVIS EVO stent represents a new advancement in the field of IA treatment. It is a self-expandable stent created specifically for application in the cerebral vasculature. The stent is made up of an inner platinum wire and an outer nitinol wire, woven together using drawn filled tube (DFT) technology. DFT enhances its visibility under fluoroscopy. This feature, along with its flexibility and short flared ends, makes it particularly suitable for use in complex aneurysm cases [21]. Additionally, it has proven effective in wide-necked aneurysms due to its high metal coverage (28%), small cell size, enhanced radiopacity, and excellent flexibility. Furthermore, it offers flow-diverting properties, making it a hybrid device suitable for both coiling assistance and standalone treatment of certain aneurysms. Several studies have reported its off-label use as a low-profile flow diverter in small and distal aneurysms [11,20,21].

The stent’s shorter flared ends and improved flexibility enable safe deployment in tortuous vessels. The most commonly treated site was the MCA (34.8%), followed by the AComA (19.6%), consistent with the higher incidence of anterior circulation aneurysms in endovascular practice [11]. Posterior circulation aneurysms are more challenging to treat due to complex vascular architecture and higher rupture rates [30]. The LVIS EVO has been successfully used in selective posterior circulation aneurysm cases, including emergency presentations, though technical demands are greater and outcomes more variable [11].

Compared to earlier versions, such as the original LVIS [31,32], LVIS Jr [32], and LEO+ Baby [33]. The LVIS EVO offers enhanced navigability and visibility, with improved apposition even in sharply angled vessels. These advancements make it more suitable for complex, bifurcation, and distal IAs than its predecessors [12]. The natural course of unruptured cerebral aneurysms in a Japanese study highlighted the higher rupture rates for aneurysms ≥ 7 mm or those located in the posterior circulation [34,35]. In our review, most aneurysms were small to medium-sized, making the LVIS EVO a favorable option due to its versatility in such cases. Its dual capacity for coil assistance and flow-diversion effect offers a safer and more adaptable alternative compared to high-density flow diverters [26].

Cautious selection becomes critical in patients with ruptured aneurysms, who may require external ventricular drainage (EVD) or other surgical interventions. In such cases, individualized dual antiplatelet therapy (DAPT) protocols and multidisciplinary planning are vital. In contrast, unruptured aneurysms allow for optimal pre-procedural planning, DAPT administration, and individualized device selection, translating into higher technical success rates and fewer complications [36]. Unlike laser-cut stents, the semi-compliant nature of LVIS EVO allows it to conform more readily to tortuous anatomies and facilitates deployment in distal or small vessels [26]. In such anatomies, deployment techniques such as the shelf technique can improve procedural outcomes and safety profile [18].

Recurrent aneurysms often feature distorted anatomy, coil compaction, or incomplete initial occlusion, making treatment more technically demanding [37]. In the context of recurrent IA retreatment, the LVIS EVO stent is a valuable tool, particularly for those previously treated with coiling [12,22], intrasaccular flow disruptor [25], or balloon-assisted coiling [23]. The LVIS EVO’s braided mesh structure allows for precise deployment over existing coil masses and improved flow remodeling [21]. These findings are further supported by Aydin et al. (2022) [25] (over 20% of cases were previously coiled IAs) and Mosimann et al. (2022) [19] (23% with prior coiling and 2% with prior clipping), collectively affirming its feasibility and safety in such complex scenarios.

### 4.3. Study Strengths and Limitations

To the best of our knowledge, this is the first systematic review and meta-analysis to provide a comprehensive assessment of the technical and clinical implications of the LVIS EVO stent in the management of IAs, offering valuable insight into this promising treatment approach. There are several limitations that warrant consideration. First, the included studies were retrospective studies, introducing potential selection bias and limiting causality. Variable definitions of complications such as thromboembolic events, in-stent stenosis, and delayed ischemic or hemorrhagic events may influence the generalizability of our safety outcomes and potentially contribute to over- or underestimation of the clinical complication rate. Additionally, some studies focused exclusively on radiographic outcomes, which may contribute to underestimation of procedure-related clinical burden. Many studies employed different closure techniques and angiographic follow-up modalities, with additional variability in imaging timing further impacting the reliability of occlusion and recurrence assessments. Due to insufficient and inconsistent reporting methods, stratified analyses by aneurysm location, rupture status, treatment setting, treatment intent, and antiplatelet regimens were not feasible, potentially limiting the generalizability across anterior and posterior circulation aneurysms. The implementation period for several included studies overlapped with the COVID-19 pandemic, which may have influenced follow-up adherence, procedural prioritization, and access to care, though this was not consistently addressed in the literature. Finally, the absence of randomized controlled trials (RCTs) and the possibility of publication bias favoring favorable outcomes reduce the overall strength of the evidence.

## 5. Future Directions

Future research should aim to address current gaps by conducting prospective, multicenter RCTs comparing LVIS EVO with other stents and flow diverters, particularly in anatomically complex or high-risk aneurysm subgroups. Standardization of imaging follow-up protocols, occlusion grading systems, and clinical outcome reporting is essential to ensure comparability across studies and to enable high-quality meta-analyses. Moreover, investigations should focus on long-term outcomes, including late recurrence rates, stent patency, and the incidence of delayed thromboembolic or hemorrhagic events. Subgroup analyses by cerebral location, vessel size, and tortuosity, as well as patient-specific factors, will enhance personalized treatment planning. The role of LVIS EVO in emergency settings, its use in distal or retreatment cases, and optimization of antiplatelet regimens remain critical areas for future exploration.

## 6. Conclusions

The LVIS EVO stent represents a promising tool in the SAC of wide-necked and morphologically complex IAs. The current evidence demonstrates high occlusion rates and low complication rates, suggesting a favorable procedural safety and efficacy profile. However, given the methodological limitations of existing studies and the lack of long-term data, further prospective, standardized, and randomized investigations are essential to confirm these findings and define optimal patient selection criteria.

## Figures and Tables

**Figure 1 jcm-15-00260-f001:**
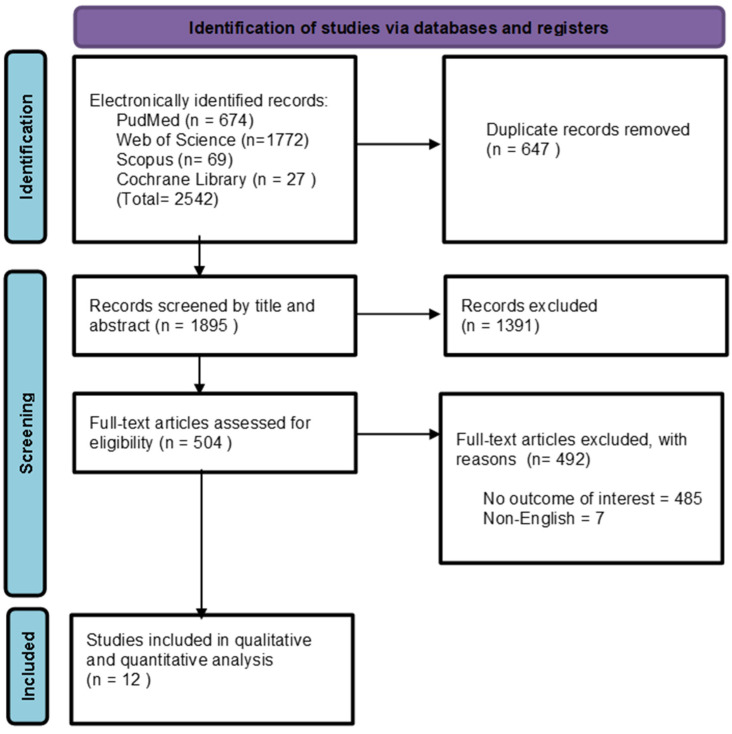
PRISMA flow diagram of the searching and screening process.

## Data Availability

All data supporting the findings of this study are available within the paper and its Appendix A.

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
