# Peer review of "Safety and Efficacy of the LVIS EVO Device for Stent-Assisted Coiling of Intracranial Aneurysms: A Systematic Review and Meta-Analysis"

_jcm, 2025, doi:10.3390/jcm15010260_

Round 1

Reviewer 1 Report

Comments and Suggestions for Authors

The authors present a timely and comprehensive systematic review and meta-analysis addressing the safety and efficacy of the LVIS EVO device for stent-assisted coiling of intracranial aneurysms. The topic is clinically relevant, and the manuscript is generally well structured and clearly written.

However, I believe that several important methodological and conceptual issues should be addressed to strengthen the scientific rigor and clinical interpretability of the findings. My major comments are outlined below.

  1. I recommend that the authors further explore possible contributors to heterogeneity, such as aneurysm rupture status, anterior versus posterior circulation, retreatment cases, or differences in treatment intent (conventional stent-assisted coiling versus flow-modifying use of LVIS EVO), through subgroup analyses or meta-regression if feasible. If additional analyses cannot be performed, this limitation should be more explicitly acknowledged and discussed, and the conclusions should be interpreted with appropriate caution.
  2. Clarification is needed as to whether the included studies distinguished between these different treatment strategies. If such differentiation was not possible, the authors should explicitly acknowledge this conceptual heterogeneity and discuss its potential impact on angiographic and clinical outcomes.
  3. I encourage the authors to expand the discussion on how heterogeneity in complication definitions and follow-up protocols may have influenced the pooled safety estimates. Emphasizing the need for standardized complication reporting in future studies would further enhance the clinical relevance of this work.

Author Response

Comment 1: [I recommend that the authors further explore possible contributors to heterogeneity, such as aneurysm rupture status, anterior versus posterior circulation, retreatment cases, or differences in treatment intent (conventional stent-assisted coiling versus flow-modifying use of LVIS EVO), through subgroup analyses or meta-regression if feasible. If additional analyses cannot be performed, this limitation should be more explicitly acknowledged and discussed, and the conclusions should be interpreted with appropriate caution.]

Response 1: [Thank you for your valuable feedback. Please find the addressed comment on pages 15–16, lines 344–355.]

Comment 2: [Clarification is needed as to whether the included studies distinguished between these different treatment strategies. If such differentiation was not possible, the authors should explicitly acknowledge this conceptual heterogeneity and discuss its potential impact on angiographic and clinical outcomes.]

Response 2: [Thank you for your valuable feedback. Please find the addressed comment on pages 15–16, lines 344–355. Please note that these analyses were not feasible, as the included studies provided only pooled data without subgrouping.]

Comment 3: [ I encourage the authors to expand the discussion on how heterogeneity in complication definitions and follow-up protocols may have influenced the pooled safety estimates. Emphasizing the need for standardized complication reporting in future studies would further enhance the clinical relevance of this work.]

Response 3: [Thank you for your valuable feedback. Please find the addressed comment on pages 15–16, lines 344–355.]

Reviewer 2 Report

Comments and Suggestions for Authors

Dear Authors,

Thank you for submitting your work. The search and study selection look solid. I read with interest. I did not find major problems with the statistics or the main results. My main concern is the topic and novelty. Your question, methods, and conclusions are very similar to a meta-analysis published earlier this year in Neurosurgery (“Technical Success and Clinical Outcomes of the Low-Profile Visualized Intraluminal Support EVO (LVIS EVO) Stent in the Treatment of Intracranial Aneurysms: A Systematic Review and Meta-Analysis”). Because of this, it is not clear what your paper adds.

Please compare your study with that paper and other similar reviews, and clearly state what is new in your work.

Author Response

Comment 1: [Please compare your study with that paper and other similar reviews, and clearly state what is new in your work.]

Response 1: [Thank you for your valuable feedback. Please find the addressed comment on lines 274–277, page 14.]

Reviewer 3 Report

Comments and Suggestions for Authors

The systematic review and meta-analysis evaluates the safety and efficacy of the LVIS EVO stent for stent-assisted coiling of intracranial aneurysms. Across 12 retrospective studies involving 567 patients, the device demonstrated high rates of aneurysm occlusion with low procedural and clinical complication rates. Despite heterogeneity and limitations inherent to non-comparative retrospective data, the findings support LVIS EVO as a safe and effective option for treating wide-necked and complex aneurysms, while highlighting the need for prospective comparative studies.

  1.  Substantial heterogeneity was observed in several key analyses (e.g., immediate RROC I and III). Can the authors elaborate on potential clinical or methodological sources of this heterogeneity beyond the leave-one-out sensitivity analysis?
  2. Why were subgroup analyses (e.g., ruptured vs. unruptured aneurysms, anterior vs. posterior circulation, primary vs. retreatment cases) not performed, and how might this affect the generalizability of the results?
  3. Follow-up duration varied considerably across included studies. How did the authors address this variability when pooling follow-up occlusion and complication rates?
  4. Antiplatelet regimens were not systematically analyzed. Given their importance in stent-assisted coiling, how might variability in dual antiplatelet therapy influence the reported thromboembolic complication rates?
  5. How do the authors envision LVIS EVO being positioned relative to other low-profile stents and flow diverters in future randomized or comparative studies?

Author Response

Comment 1: [Substantial heterogeneity was observed in several key analyses (e.g., immediate RROC I and III). Can the authors elaborate on potential clinical or methodological sources of this heterogeneity beyond the leave-one-out sensitivity analysis?]

Response 1: [Thank you for your valuable feedback. Please find the addressed comment on lines 271–273, page 14 and pages 15-16 lines 344-355]

Comment 2: [Why were subgroup analyses (e.g., ruptured vs. unruptured aneurysms, anterior vs. posterior circulation, primary vs. retreatment cases) not performed, and how might this affect the generalizability of the results?]

Response 2: [Thank you for your valuable feedback. Please find the addressed comment on pages 15-16 lines 344-355]

Comment 3: [Follow-up duration varied considerably across included studies. How did the authors address this variability when pooling follow-up occlusion and complication rates?]

Response 3: [Thank you for your valuable feedback. Following well-established and widely accepted guidelines for systematic reviews and meta-analyses, follow-up data were extracted as reported at the last documented time point for each study. We acknowledge that heterogeneity may arise from differences in follow-up duration and assessment tools, as noted in our previous comments.]

Comment 4: [Antiplatelet regimens were not systematically analyzed. Given their importance in stent-assisted coiling, how might variability in dual antiplatelet therapy influence the reported thromboembolic complication rates?]

Response 4: [Thank you for your valuable feedback. Please find the addressed comment on pages 15-16 lines 344-355]

Comment 5: [How do the authors envision LVIS EVO being positioned relative to other low-profile stents and flow diverters in future randomized or comparative studies?]

Response 5: [Thank you for your valuable feedback. Throughout the Discussion, we aimed to provide evidence-based information on this promising stent. Given the observational nature of the included studies, it is difficult to draw conclusions beyond the well-documented outcomes reported in the literature. Therefore, we have provided rational interpretations and commentary throughout the Discussion, while emphasizing the need for future randomized controlled trials and additional studies to assess the long-term reliability and value of the LVIS EVO stent.]

Reviewer 4 Report

Comments and Suggestions for Authors

Authors need to use the journal's citation format. See more comments in the attached file. 

It adheres to PRISMA guidelines and was registered with PROSPERO, so it is a formal research.

Author Response

Comment 1: [Authors need to use the journal's citation format. See more comments in the attached file.]

Response 1: [Thank you for your valuable feedback. According to the journal’s free-style submission guidelines, we submitted our manuscript using the implemented citation style. The production team will assist in rearranging the references according to the final proofreading version.

Furthermore, we have refined the abbreviations based on your comments.

Regarding the tables, we agree that some formatting adjustments are needed. We have previously contacted the editor to address this, as table formatting cannot be changed on our side due to the journal’s submission style. We trust that the production team will finalize the formatting after acceptance.]

Round 2

Reviewer 1 Report

Comments and Suggestions for Authors

Thank you for your careful revisions to the manuscript in response to my comments.